# Reasons behind the Delayed Diagnosis of Testicular Cancer: A Retrospective Analysis

**DOI:** 10.3390/ijerph20064752

**Published:** 2023-03-08

**Authors:** Wojciech A. Cieślikowski, Michał Kasperczak, Tomasz Milecki, Andrzej Antczak

**Affiliations:** Department of Urology, Poznan University of Medical Sciences, 61-701 Poznan, Poland

**Keywords:** testicular cancer, diagnosis, survival, screening, public health

## Abstract

The aim of the present study was to identify the reasons behind the delayed diagnosis of testicular cancer in a group of Polish males diagnosed with this malignancy in 2015–2016. The study included data from 72 patients aged between 18 and 69 years. Based on the median time elapsed to the testicular cancer diagnosis, the study patients were divided into the timely diagnosis group (diagnosis within 10 weeks from initial manifestation, *n* = 40) and the delayed diagnosis group (diagnosis > 10 weeks from initial manifestation, *n* = 32). Diagnosis of testicular cancer > 10 weeks after its initial manifestation was associated with less favorable survival (5-year overall survival: 78.1% [95% CI: 59.5–88.9%] vs. 92.5% [95% CI: 78.5–97.5%], *p* = 0.087). Multivariate logistic regression analysis identified two independent predictors of the delayed diagnosis, age > 33 years (OR = 6.65, *p* = 0.020) and residence in the countryside (OR = 7.21, *p* = 0.012), with another two parameters, the lack of a regular intimate partner (OR = 3.32, *p* = 0.098) and the feeling of shame (OR = 8.13, *p* = 0.056), being at the verge of statistical significance. All the factors mentioned above should be considered during planning social campaigns aimed at the early detection of testicular malignancies, along with improving the quality and trustfulness of Internet-based information resources.

## 1. Introduction

Testicular cancer is the most common malignancy diagnosed in younger males (20–44 years of age), with the yearly number of new cases estimated at 71,000 worldwide and 1200 in Poland [1,2]. Usually, the disease is diagnosed at early stages, and with radical orchidectomy, 5-year cancer-specific survival rates are high, up to 95% [3]. However, the prognosis worsens considerably if testicular cancer is diagnosed at higher clinical stages [4]. The data included in the Surveillance, Epidemiology, and End Results (SEER) program database created by the US National Cancer Institute show clearly that the stage at the diagnosis had a dramatic impact on prognosis in American patients diagnosed with testicular cancer between 2011 and 2017. While the 5-year relative survival rate for patients diagnosed with localized testicular cancer, limited to the testicles, approximated 99%, the likelihood of survival decreased to 96% in the case of regional malignancies involving regional lymph nodes and surrounding structures and down to 73% if distant metastases occurred [5].

Like in the case of other malignancies, a key to the early diagnosis of testicular cancer is the awareness of its early manifestations and appropriate screening [6]. Typically, testicular cancer manifests as painless enlargement of the testicle or the presence of a lump of variable size. Alternatively, the first manifestation of the malignancy can be pain, discomfort, or numbness within the testicle or entire scrotum, with accompanying swelling or without. Some patients note changes in the structure of one testicle, which appears firmer or ‘heavier’ than the other. Others may report an accumulation of fluid within the scrotum. Given the nature of testicular cancer’s manifestations mentioned above, testicular self-examination is the key to early diagnosis of this malignancy. The examination, including visual inspection followed by careful palpation of each testicle with both hands, should be performed approximately once a month by all males aged 15 to 45. A patient in whom testicular self-examination demonstrated any of the abovementioned abnormalities should seek medical consultation from a urologist as soon as possible [7]. If a patient presents with a suspected mass within the scrotum, the test of choice is the ultrasonographic examination of the area. During a scrotal ultrasound, the physician determines the diameter and location of the mass and verifies whether it is located within the testicle or separate within the scrotum, solid, or filled with fluid. Usually, malignant lesions present as solid lumps located within the testicle. Additionally, tumor markers are determined within the patient’s blood, including beta-human chorionic gonadotropin, alpha-fetoprotein, and lactate dehydrogenase. If the results of scrotal ultrasound and laboratory testing indicate that the lesion is likely malignant, the diagnosis is ultimately verified histopathologically after removing the affected testicle. Additionally, imaging studies are conducted to detect potential distant metastases if justified by a clinical presentation [7].

Importantly, testicular cancer, a malignancy occurring at a relatively younger age, has a detrimental effect on male reproductive health. According to the literature, more than 50% of men diagnosed with testicular cancer present with oligospermia already before the implementation of any treatment. While the chances of fathering a child after purely surgical treatment of the malignancy are relatively higher (90%), the probability of conceiving a baby by patients who received adjuvant chemotherapy after being diagnosed with late-stage testicular cancer decreases dramatically, down to 48%. Hence, in many testicular cancer survivors, having offspring will require assisted reproductive technology, which further increases the economic burden of the disease [8].

A few previous studies demonstrated that the knowledge of the early symptoms of testicular cancer among males at risk is suboptimal [9,10,11,12,13], with the practice of testicular self-examination, a basic screening test, not being as common as recommended [14,15,16,17,18,19]. However, still little is known about the causes of diagnostic delay in men with testicular malignancies. Some authors pointed to poor socioeconomic status as an obstacle in seeking medical advice early [20,21], but this explanation is not necessarily applicable to men from developed countries having unrestricted access to healthcare resources.

The aim of the present study was to identify the reasons behind the delayed diagnosis of testicular cancer in a group of Polish males.

## 2. Materials and Methods

### 2.1. Participants

The retrospective analysis included data of all consecutive patients operated on for testicular cancer at the Department of Urology, Poznan University of Medical Sciences (Poznan, Poland) in 2015–2016. All patients were referred by local practitioners with an initial diagnosis/suspicion of testicular cancer, and the ultimate diagnosis was established based on testicular ultrasound and tumor marker determination. The treatment consisted of inguinal orchidectomy, followed by adjuvant treatment in eligible cases [22]. All patients were routinely followed up for five years, in line with the European Association of Urology guidelines [22].

At the time of initial evaluation, as a part of the routine admission procedure, all patients completed a structured interview, including questions about their demographics, initial manifestations of testicular malignancy, available sources of knowledge about testicular cancer, and obstacles in seeking medical advice early.

### 2.2. Statistical Analysis

Data from the survey were subjected to statistical analysis as potential determinants of delayed diagnosis. The chi-square and the rank-sum tests were used to compare categorical and continuous variables, respectively. Logistic regression analysis was carried out to identify significant predictors of the diagnostic delay, with the results reported as odds ratios (OS) with 95% confidence intervals (CI). Variables with significant *p*-values in the univariate analysis were then included in the multivariate analysis. For the survival analysis, the Kaplan–Meier method was used to generate overall survival (OS) curves. All reported *p*-values are two-sided and were considered significant if less than 0.05. Calculations and graphics were obtained using STATA IC 16.1 (StataCorp, College Station, TX, USA).

## 3. Results

### 3.1. General Characteristics of the Study Participants

The analysis included the data of 72 male patients aged between 18 and 69 years (mean 33.9 ± 10.4 years, median 33 years). The time elapsed from the initial manifestation to the ultimate diagnosis of testicular cancer varied between 3 and 100 weeks (median 10 weeks). Surgical treatment was implemented within 1 to 4 weeks (median 1 week) after establishing the diagnosis.

### 3.2. Determinants of Delayed Diagnosis

Based on the median time elapsed to the testicular cancer diagnosis, the study patients were divided into the timely diagnosis group (diagnosis within 10 weeks from initial manifestation, *n* = 40) and the delayed diagnosis group (diagnosis > 10 weeks from initial manifestation, *n* = 32). Detailed characteristics of those groups are shown in Table 1.

In the delayed diagnosis group, painless enlargement of the testicle was the initial manifestation of cancer significantly more often than in the timely diagnosis group (69% vs. 22%, *p* < 0.001). In contrast, pain or the presence of a palpable nodule within the testicle were reported significantly less frequently (12% vs. 37%, *p* = 0.015 and 19% vs. 42%, *p* = 0.028, respectively). Compared with participants from the timely diagnosis group, patients from the delayed diagnosis group significantly more often resided in the countryside (59% vs. 20%, *p* = 0.003) and completed solely elementary education (62% vs. 17%, *p* < 0.001). Moreover, they significantly less often had a regular intimate partner (spouse, girlfriend) than those from the timely diagnosis group (34% vs. 75%, *p* = 0.001). When asked about the sources of their knowledge about testicular cancer, patients from the delayed diagnosis group mentioned Internet resources significantly more often than their timely diagnosis counterparts (66% vs. 30%, *p* = 0.003). Meanwhile, persons from the timely diagnosis group obtained information about testicular cancer from a urologist significantly more often than those from the delayed diagnosis group (45% vs. 6%, *p* < 0.001). When surveyed about the reasons behind their late referral to the specialist, patients from the delayed diagnosis group significantly more often than those from the timely diagnosis group mentioned the feeling of shame (31% vs. 5%, *p* = 0.004) and significantly less often pointed to the lack of adequate knowledge/negligence of symptoms (56% vs. 85%, *p* = 0.007).

During the next step, logistic regression analysis was carried out to identify the independent predictors of the delayed diagnosis. The logistic regression models included elementary education, residence in the countryside, lack of a regular intimate partner, the Internet as a primary source of information about testicular cancer, and the feeling of shame. All those variables turned out to be highly significant predictors of delayed diagnosis on univariate analysis. However, multivariate analysis, including all those variables, along with patient age, identified only two independent predictors, age > 33 years (OR = 6.65, 95% CI 1.34–32.98, *p* = 0.020) and residence in the countryside (OR = 7.21, 95% CI: 1.54–33.72, *p* = 0.012), with another two parameters, the lack of a regular intimate partner (OR = 3.32, 95% CI: 0.80–13.72, *p* = 0.098) and the feeling of shame (OR = 8.13, 95% CI: 0.95–69.59, *p* = 0.056), being at the verge of statistical significance (Table 2).

To better understand the role of the two independent predictors identified above, we analyzed their relationships with the causes of diagnostic delay mentioned in the survey (Table 3). Patients residing in the countryside significantly more often than those living in larger municipalities declared the feeling of shame (30% vs. 9%, *p* = 0.026) and significantly less often pointed to the lack of adequate knowledge/negligence of symptoms as a reason behind the diagnostic delay (52% vs. 84%, *p* = 0.004). No significant differences in the accessibility of medical care were found between the countryside and larger municipality dwellers. The lack of adequate knowledge/negligence of symptoms was significantly more often mentioned as a factor responsible for the diagnostic delay by patients older than 33 years than by younger persons (87% vs. 60%, *p* = 0.009). Meanwhile, younger patients mentioned the feeling of shame significantly more often than those aged > 33 years (27% vs. 3%, *p* = 0.005). The two age groups did not differ significantly regarding the declared availability of healthcare resources.

### 3.3. Effect of Delayed Diagnosis on Clinical Stage and Treatment Outcome

Patients from the timely diagnosis group differed, at the verge of statistical significance, from those from the delayed group in terms of the distribution of pT stages at the time of diagnosis, with the latter being significantly more often diagnosed with pT2 or pT3 tumors than the former (pT2: 34% vs. 22%, pT3: 25% vs. 10%, *p* = 0.081).

A total of 10 patients died during a 5-year follow-up period (median survival: not reached, 5-year OS: 86.1% [95% CI: 75.7–92.3%]; Figure 1a): among them, 7 from the delayed diagnosis group and 3 from the timely diagnosis group (median survival: not reached in either group, 5-year OS: 78.1% [95% CI: 59.5–88.9%] vs. 92.5% [95% CI: 78.5–97.5%], *p* = 0.087). Diagnosis of testicular cancer >10 weeks after its initial manifestation was associated with less favorable survival (Figure 1b).

## 4. Discussion

The present study identified two factors, age > 33 years and living in the countryside, as the independent predictors of delayed diagnosis of testicular cancer. Another two factors, the lack of a regular intimate partner and the feeling of shame, were shown to be closely related to the diagnostic delay, although none of them reached the threshold of statistical significance on multivariate logistic regression analysis. Not surprisingly, the delayed diagnosis turned out to be associated with worse OS in testicular cancer.

The further in-depth analysis demonstrated that respondents older than 33 years significantly more often than the younger participants mentioned the lack of adequate knowledge/negligence of symptoms as a reason for the diagnostic delay. Males aged > 33 years are usually occupationally active and have their own families. Published evidence suggests that being overloaded with both job- and family-related duties, such persons frequently postpone referral to a physician despite the presence of potentially alarming symptoms; this can lead to a diagnostic delay [23]. Moreover, occupationally active men may not have enough time to actively seek information about their developed symptoms [24,25]. Finally, some men, especially older ones, may postpone the referral to a physician because of carcinophobia, preferring a passive attitude over receiving a devastating diagnosis [26]. Meanwhile, the results of our analysis suggest that the respondents ≤ 33 years of age more often than the older patients possessed adequate knowledge about the symptoms of testicular cancer and did not ignore initial manifestations of the disease. This might be associated with the fact that younger persons, frequently without a permanent intimate partner, generally tend to be more concerned about their sexual health [27]. However, our analysis also showed that despite being aware of the disease and its symptoms, younger respondents postponed their visit to a physician due to shame, a problem discussed in more detail below.

Aside from older age, living in the countryside was another significant determinant of delayed diagnosis. One would expect the limited availability of healthcare resources as the main reason for the delayed diagnosis among countryside dwellers, consistent with some published data [28]. However, the results of the present study suggest otherwise. Our analysis demonstrated that the primary cause of the delayed diagnosis in males residing in the countryside was the feeling of shame. The embarrassment associated with referral to a physician because of ‘male problems’ seems to be a problem in highly religious Polish society, where sex and sexual organs are still a kind of taboo. Furthermore, it should be remembered that people in small communities know each other well and gossip frequently. In contrast, residence in a larger town or a big city allows the patient to visit a physician without being concerned that information about his health problems would be shared with others; this sense of anonymity explains why our respondents living in larger municipalities identified the feeling of shame as a reason for diagnostic delay significantly less often than the countryside dwellers.

The results of the present study highlighted an essential role of a spouse/intimate partner in earlier diagnosis of testicular cancer. Many previous studies demonstrated that having a permanent intimate partner can constitute a significant determinant of timely diagnosis in both male and female malignancies [29,30,31,32,33,34]. This fact is worth emphasizing, given that one symptom more common in the delayed diagnosis group than in the timely diagnosis group was a painless enlargement of the testicle. The enlargement of the testicle without accompanying pain may be easily overlooked by the patient but not necessarily by his partner.

Importantly, patients from the delayed diagnosis group significantly more often than their timely diagnosis counterparts mentioned the Internet as a primary source of information about testicular cancer. This finding points to a growing problem of seeking medical information from unverified Internet resources instead of referring to a specialist [9,10,11,12,13]. This seems to be particularly widespread in the case of medical problems related to sexual health. One potential solution is the development of trusted Internet resources for patients affiliated with renowned clinics or scientific bodies. Another option is the creation of social campaigns and awareness programs oriented at promoting testicular self-examination [14,15,16,17,18,19] and early detection of testicular malignancies [35,36,37,38]. Examples of such initiatives are Testicular Cancer Awareness Month and Men’s Health Awareness Month, held every April and November, respectively.

In the present study, the delayed diagnosis of testicular cancer had an unfavorable impact on the clinical stage of the malignancy, which in turn negatively affected the OS. Diagnostic delay is an established determinant of unfavorable outcomes in testicular cancer and other malignancies [6,39,40]. This fact highlights the importance of building cancer awareness in the general population; one of the ways to achieve this goal is by overcoming psychosocial obstacles for timely diagnosis, such as those identified in the present study.

To summarize, the results of the present study show clearly that the knowledge of testicular cancer symptoms among Polish males is insufficient, with men with some specific sociodemographic backgrounds presenting more severe gaps in this matter than others. One key to addressing the problem in question is to popularize the practice of testicular self-examination. Only by being familiarized with the normal appearance of their testicles at an early age and self-examining them regularly every month can males detect potential abnormalities early and seek medical advice promptly. Aside from being aware of the early manifestations of testicular cancer, males should also realize the potential consequences of detecting the malignancy at more advanced stages. These include not only markedly lower chances of survival in patients in whom testicular cancer was diagnosed at the stage when systemic spread had already occurred but also a detrimental effect of systemic anticancer treatment on fertility. Only strong healthcare awareness combined with adequate access to healthcare resources may prevent diagnostic delay in testicular cancer and the consequences thereof, not only clinical but also economic ones.

This study has some potential limitations. First, this was a single-center study with a relatively small sample size. Thus, the results presented herein should be verified in a larger, population-based study. A larger study, including male respondents with more variable sociodemographic backgrounds, would allow us to verify the findings presented herein and perhaps identify some additional factors contributing to the delayed diagnosis of testicular cancer. Second, we retrospectively analyzed available data included in the survey and hence were unable to consider some other well-known determinants of diagnostic delay in cancer patients, such as economic status [4,21]. Third, analyzing the outcomes, we relied on the respondents’ subjective declarations instead of verifying their cancer awareness and psychological disposition with validated instruments.

## 5. Conclusions

Timely diagnosis of testicular cancer can be delayed due to the older age of the patient, residence in the countryside, lack of an intimate partner, and/or feeling of shame. All those factors should be considered during planning social campaigns aimed at the early detection of testicular malignancies, along with improving the quality and trustfulness of Internet-based information resources.

## Figures and Tables

**Figure 1 ijerph-20-04752-f001:**
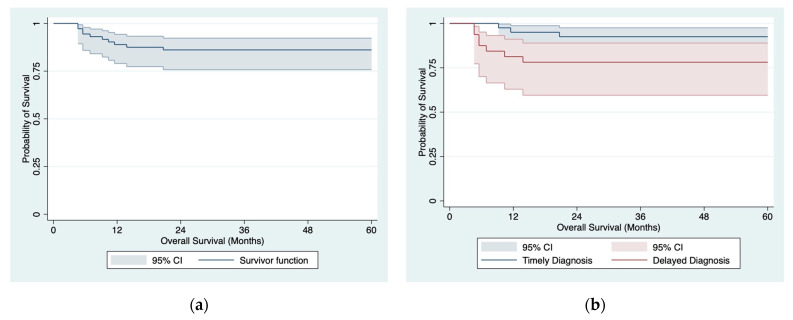
OS estimates in 72 patients with testicular cancer: (**a**) entire cohort; (**b**) by the time of diagnosis.

**Table 1 ijerph-20-04752-t001:** Summary of the survey results for testicular cancer patients in whom the malignancy has been diagnosed timely and those with a diagnostic delay.

Variable	Delayed Diagnosis (>10 Weeks, *n* = 32)	Timely Diagnosis (≤10 Weeks, *n* = 40)	*p*-Value
Age in years (mean ± SD)	34.4 ± 11.7	33.4 ± 9.4	0.966
Education, *n* (%)			
-elementary	20 (62%)	7 (17%)	<0.001
-secondary	12 (37%)	14 (35%)	
-higher	0 (0%)	19 (47%)	
Place of residence, *n* (%)			
-big city	8 (25%)	18 (45%)	0.003
-larger/mid-sized town	5 (16%)	14 (35%)	
-countryside	19 (59%)	8 (20%)	
Regular intimate partner, *n* (%)	11 (34%)	30 (75%)	0.001
Initial manifestation, *n* (%)			
-painless enlargement	22 (69%)	9 (22%)	<0.001
-pain	4 (12%)	15 (37%)	0.015
-palpable nodule	6 (19%)	17 (42%)	0.028
Source of information, *n* (%)			
-intimate partner	0 (0%)	1 (2%)	0.556
-Internet	21 (66%)	12 (30%)	0.003
-family physician	10 (31%)	15 (37%)	0.382
-urologist	2 (6%)	18 (45%)	<0.001
Reason for delayed referral, *n* (%)			
-negligence of symptoms	18 (56%)	34 (85%)	0.007
-shame	10 (31%)	2 (5%)	0.004
-limited access to healthcare	3 (9%)	2 (5%)	0.394
-wrong initial diagnosis	2 (6%)	2 (5%)	0.604

**Table 2 ijerph-20-04752-t002:** Factors associated with the delayed diagnosis of testicular cancer: the results of logistic regression analysis.

Explanatory Variable	Univariate Analysis		Multivariate Analysis	
	OR (95% CI)	*p*-Value	OR (95% CI)	*p*-Value
Age > 33 years	1.19 (0.47–3.04)	0.711	6.65 (1.34–32.98)	0.020
Elementary education	7.86 (2.65–23.25)	<0.001	2.53 (0.54–11.85)	0.239
Countryside residence	5.85 (2.05–16.67)	0.001	7.21 (1.54–33.72)	0.012
Lack of regular partner	5.73 (2.06–15.91)	0.001	3.32 (0.80–13.72)	0.098
Information from Internet	4.45 (1.65–12.04)	0.003	1.32 (0.34–5.12)	0.687
Shame	8.64 (1.73–43.05)	0.009	8.13 (0.95–69.59)	0.056

**Table 3 ijerph-20-04752-t003:** Relationships between participants’ demographics and causes of delayed diagnosis in testicular cancer: a cross-sectional analysis.

Place of Residence			
Reason for delayed referral:	Countryside (*n* = 27)	Other (*n* = 45)	*p*-value
-negligence of symptoms	14 (52%)	38 (84%)	0.004
-shame	8 (30%)	4 (9%)	0.026
-limited access to healthcare	2 (7%)	3 (7%)	0.625
-wrong initial diagnosis	3 (11%)	1 (2%)	0.145
**Age**			
Reason for delayed referral:	>33 years (*n* = 32)	≤33 years (*n* = 40)	*p*-value
-negligence of symptoms	28 (87%)	24 (60%)	0.009
-shame	1 (3%)	11 (27%)	0.005
-limited access to healthcare	3 (9%)	2 (5%)	0.394
-wrong initial diagnosis	0 (0%)	4 (10%)	0.089

## Data Availability

The datasets generated and/or analyzed during the current study are not publicly available because of patient privacy and legal and administrative policies of the medical institution where the study was conducted but are available from the corresponding author upon reasonable request.

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
