# Peer review of "Reasons behind the Delayed Diagnosis of Testicular Cancer: A Retrospective Analysis"

_ijerph, 2023, doi:10.3390/ijerph20064752_

Round 1

Reviewer 1 Report

I appreciate reviewing this paper. I will keep my comments more general instead of minute grammatical changes.

1) Address the small sample size. This could affect your findings greatly and I think you need more discussion on this fact.

2) You need more conversation on "take homes" for the readers. What exactly can we do with this information? I think you need to get more specific 

Author Response

Thank you for your valuable comments. They have been all addressed in the revised manuscript.

I appreciate reviewing this paper. I will keep my comments more general instead of minute grammatical changes.

Re: The manuscript has been revised for grammar and style by an English speaker experienced in revising medical articles.

Address the small sample size. This could affect your findings greatly and I think you need more discussion on this fact.

Re: This issue has been addressed in the revised limitations paragraph of the Discussion.

You need more conversation on "take homes" for the readers. What exactly can we do with this information? I think you need to get more specific 

Re: A separate paragraph with the take-home message has been added to the revised Discussion.

Reviewer 2 Report

In the manuscript, entitled "Reasons Behind the Delayed Diagnosis of Testicular Cancer: a 2 Retrospective Analysis”, the authors of this study are trying to investigate the basic reasons for the delayed detection of testicular cancer clinically. 

Comments :

1. It would be very helpful if the authors do provide the world and Poland statistics data on Testicular Cancer. (More information)

2. Please provide the details of symptoms of Testicular Cancer in the introduction section.

3.Please provide the implications of Testicular Cancer on male physiology/reproductive health in the introduction section.

4.Please provide what diagnosis methods have been identified till today for Testicular Cancer.

5. Please also explain briefly why the early diagnosis is so critical for the patient.

6.Please remove the separate heading of the survey and merged it with the participants.

7.It would be recommended to refine the language throughout the manuscript before publishing.

Author Response

Thank you for your valuable comments. They have been all addressed in the revised manuscript.

It would be very helpful if the authors do provide the world and Poland statistics data on Testicular Cancer. (More information)

Re: As per your suggestion, the most recent Polish epidemiological data have been added to the Introduction.

Please provide the details of symptoms of Testicular Cancer in the introduction section.

Re: A separate paragraph addressing this problem has been added to the Introduction.

Please provide the implications of Testicular Cancer on male physiology/reproductive health in the introduction section.

Re: This issue has been addressed in the revised Introduction section.

Please provide what diagnosis methods have been identified till today for Testicular Cancer.

Re: A separate paragraph addressing this problem has been added to the Introduction.

Please also explain briefly why the early diagnosis is so critical for the patient.

Re: The problem has been addressed in the revised Introduction using the most recent data from the SEER database affiliated by the US National Cancer Institute.

Please remove the separate heading of the survey and merged it with the participants.

Re: The text has been rearranged as per your suggestion.

It would be recommended to refine the language throughout the manuscript before publishing.

Re: The manuscript has been checked for grammar and style by an English native-speaker experienced in revising medical texts.

Reviewer 3 Report

In this manuscript, the authors have reported results demonstrating identify the reasons behind the delayed diagnosis of testicular cancer in a group of Polish males diagnosed with this malignancy in 2015-2016. The study can be greatly improved if the following suggestions were incorporated.

1-           Please add some studies related to history of Delayed Diagnosis of Testicular Cancer in the main introduction.

2-           Line 412, sentence “In contrast, residence in a larger town or a big city provides the patient with a sense of anonymity; this explains why our respondents living in larger municipal…” and line 99 “Male camels achieve puberty at approximately three years old, but do not fully” are not meaningful. Please check the grammar.

3-           Line 126, the sentence “Published evidence suggests that being overloaded with both job- and family ..” is not cited. 

Author Response

Thank you for your valuable comments. They have been all addressed in the revised manuscript.

Please add some studies related to history of Delayed Diagnosis of Testicular Cancer in the main introduction.

Re: This issue has been addressed in the revised Introduction using the most recent survival data from the SEER database affiliated with the US National Cancer Institute.

Line 412, sentence “In contrast, residence in a larger town or a big city provides the patient with a sense of anonymity; this explains why our respondents living in larger municipal…” and line 99 “Male camels achieve puberty at approximately three years old, but do not fully” are not meaningful. Please check the grammar.

Re: The sentence in line 412 has been revised accordingly. Unfortunately, we could not find the sentence about male camels in line 99.

Line 126, the sentence “Published evidence suggests that being overloaded with both job- and family ..” is not cited. 

Re: An appropriate reference has been added to the abovementioned sentence.

Round 2

Reviewer 2 Report

Recommendation: Accepted (Satisfactory amendments after revision)